# How well do national and local policies in England relevant to maternal and child health meet the international standard for non-communicable disease prevention? A policy analysis

Daniel Penn-Newman,[1] Sarah Shaw,[1] Donna Congalton,[2] Sofia Strommer,[1] Taylor Morris,[1] Wendy Lawrence,[1,3] Debbie Chase,[4] Cyrus Cooper,[1,3] Mary Barker,[1,3] Janis Baird,[1,3] Hazel Inskip,[1,3] Christina Vogel[1,3]

For numbered affiliations see end of article.

**Correspondence to**
Dr Christina Vogel;
cv@mrc.soton.ac.uk

## ABSTRACT

**Objectives** (1) To identify national policies for England and local policies for Southampton City that are relevant to maternal and child health. (2) To quantify the extent to which these policies meet the international standards for nutrition and physical activity initiatives set out in the WHO Global Action Plan for the Prevention and Control of Non-Communicable Diseases (WHO Action Plan).

**Design** The policy appraisal process involved three steps: (1) identifying policy documents relevant to maternal and infant health, (2) developing a policy appraisal framework from the WHO Action Plan, and (3) analysing the policies using the framework.

**Setting** England and Southampton City.

**Participants** 57 national and 10 local policies.

**Results** Across both national and local policies, priority areas supporting public health processes, such as evidence-based practice, were adopted more frequently than the action-oriented areas targeting maternal and child dietary and physical activity behaviours. However, the policy option managing conflicts of interest was rarely considered in the national policies (12%), particularly in white papers or evidence-based guidelines. For the action-oriented priority areas, maternal health policy options were more frequently considered than those related to child health or strengthening health systems. Complementary feeding guidance (9%) and workforce training in empowerment skills (14%) were the least frequent action-oriented policy options adopted among the national policies. The maternal nutrition-focused and workforce development policy options were least frequent among local policies adopted in 10% or fewer. Macroenvironmental policy options tended to have a lower priority than organisational or individual options among national policies (p=0.1) but had higher priority among local policies (p=0.02).

**Conclusions** Further action is needed to manage conflicts of interest and adopt policy options that promote a system-wide approach to address non-communicable diseases caused by poor diet and physical inactivity.

## Strengths and limitations of this study

► This is the first policy review to focus on maternal, infant and child health for non-communicable disease prevention and considers government activity at both national and local levels.
► The assessment method reflects adherence to the WHO international best practice and enables easy identification of areas for improvement.
► Use of a second assessor and verification by policy makers provided quality assurance.
► Policies from only one local authority were assessed as a case study.
► The use of a dichotomous measure does not provide comprehensive details about the scale of policy implementation.

## INTRODUCTION

Non-communicable diseases (NCDs) including cardiovascular disease and diabetes are the primary cause of death globally.[1] In the UK, they account for 89% of all adult deaths.[2] Exposure to risk factors such as obesity, physical inactivity and poor diet begins early, before conception and in utero.[3] Accordingly, optimising maternal, infant and child health is a top priority on the international health agenda. There is thus a need to align the NCD prevention agenda with the high-level directives encouraging member states to enact nutrition initiatives targeting preconception, pregnant and breastfeeding mothers and the first 1000 days.[4–6]

National and local governments' policies play a key role in leading action on NCD prevention. National governments are best placed to monitor population behaviours and health indicators, and have a responsibility to enact policies that will optimise the health of

their citizens.[7] Local governments have the ability to work with the community, respond to local needs and create local settings that support healthier lifestyle behaviours, particularly in the UK where public health responsibilities were deferred to local authorities in 2012.[8] The content of local and national policies may vary considerably or may be closely aligned, depending on whether there is strong national leadership on an issue or high heterogeneity in local authorities' structures and priorities.[9] While jurisdictional responsibilities differ between government levels and only national governments can enact legislation, for example, on nutrition labelling, there are activities that local governments can adopt on the same issue, such as shelf or menu nutrition prompts in cafeterias in schools, hospitals or leisure centres.

National government policies cover a variety of document types that hold different levels of authority from clear government directive to recommending an action in a normative sense. One set of categories applicable in the UK includes: (1) acts and codes: legally binding approved by the House of Commons and Lords that are implemented by an appropriate government agency,[10] (2) white papers: an outline of government's strategic direction or priorities for action on a particular subject[11] and (3) evidence-based guidelines: implicit policies including summaries of evidence into advice on best-practice standards or recommended course of action, such as guidance by the National Institute for Health and Care Excellence (NICE).[12] Different types of national policies may vary in scope or represent comprehensive, complementary action depending on a government's political ideology, time in office and power across the parliamentary houses, among other factors.[13] Policy options to address an issue are often raised initially in policy documents with less authority, to be discussed and debated in parliament before adoption as a government directive. Assessing similarities across or differences between types of policy documents can provide insight into areas for improvement, particularly when compared with international standards.

To strengthen state members' efforts to address the growing global burden of NCDs, the 66th World Health Assembly endorsed the WHO Global Action Plan for the Prevention and Control of NCDs (Action Plan) and NCD Global Monitoring Framework in 2013.[14] The Action Plan recognises the primary role of governments in responding to the growing challenge of NCDs and highlights the links between poor diet, physical inactivity and NCDs. It also states the importance of enhancing maternal and child health. The Action Plan sets out a range of policy options that governments at all levels can adopt and provides an international standard against which government action can be assessed. The policy options work across the ecological model of health.[15] They target: (1) individual-level behaviour change, (2) organisational-level determinants including social and physical settings and (3) macroenvironmental-level determinants including fiscal policy and infrastructure. Multicomponent policy action that works across these levels has shown to be more effective at improving lifestyle behaviours than action at a single level,[16] though it is recognised that macroenvironmental policy action is politically challenging.[7]

The current NCD Global Monitoring Framework measures progress in reducing prevalence rates of health outcomes or risk factors but does not comprehensively assess progress on policy options to improve both dietary and physical activity behaviours.[17] Other policy appraisal frameworks focus on national government progress for a single behaviour and have not considered policy options specific to maternal, infant and child health.[13 18] The 2014 report of the Chief Medical Officer highlighted that half the population of women aged 25–34 years in England are overweight or obese and noted the need to improve dietary and physical activity behaviours among women of reproductive age.[19] Additionally, the public health white paper for England 'Healthy Lives Healthy People' and the 'Childhood Obesity' strategy highlight the need for action across government sectors and levels to address the persistent public health issue of obesity. Assessment of how well national and local government policies are supporting women and children to adopt healthy dietary and physical activity behaviours is needed to identify gaps for further government action. Therefore, this study aimed to quantify the extent to which English national and local government policies that are relevant to maternal and child health meet the international standard set out in the WHO Action Plan. There were three aims of this study. First, to identify national English government policies and local government policies, using Southampton City as a case study. Second, to analyse the extent of adoption of the Action Plan policy options in these policies. Third, to assess differences in adoption of the policy options by policy type (local/national and acts/white papers/evidence-based guidance) and across three levels of an ecological model of health.

## METHODOLOGY

The policy appraisal process involved three steps. The first step was to identify national (England) and local policies (Southampton City) that could influence maternal and child health. The second step involved developing a policy appraisal framework based on relevant WHO Action Plan policy options. The third step was to analyse the identified policies using the appraisal framework. Ethical approval was not required for this study because it was a secondary analysis of publicly available policy documents.

### Identification of national and local policy documents

The policies were identified in December 2015, and an updated search was conducted in June 2017. At each time point, government websites were exhaustively searched to identify acts and codes, briefing notes, white papers and evidence-based guidelines that were relevant to maternal and child health. We adopted a broad definition of maternal and child health policy by including policy

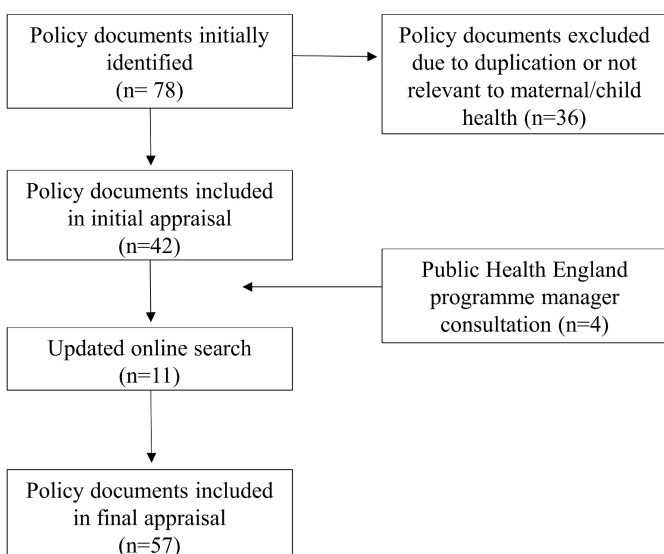

**Figure 1** The schematic diagram of the policy identification process.

documents that targeted the health of women and/or children, as well as those that could indirectly influence the health or dietary or physical activity behaviours of women and children. Accordingly, a range of websites were searched including the Department of Health, Department for Education, Department for Children Schools and Families and the Department for Communities and Local Government. Additionally, executive agency websites, including the Food Standards Agency, Public Health England, NICE and the Local Government Association, were searched because these agencies can influence government policy development. Policies focusing on disease treatment or specific nutrients or those detailed in policies published at a later date were excluded. The compilation of the identified national policies was then sent to a programme manager at Public Health England for verification and identification of missing policies. Documents published during the UK Coalition and Conservative Governments, from May 2010 to June 2017, were the focus of this review. Key policies prior to this date were included if they: (1) informed current policies or (2) had not been superseded.

Southampton City was used as a case study for the local policy analysis. Southampton is a large city on the south coast, more deprived than other cities in the affluent South East of England. Southampton is ranked the 67th most deprived of the 326 local authorities in England.[20] In 2014/2015, 47% of women in Southampton were classified as overweight or obese at their maternity booking appointment,[21] indicating that this local health issue is similar to the national and international level. To aid methodological rigour and completeness, the identification of local Southampton policies was supported and verified by a public health consultant (DeC) at the Southampton City Council. Policies were extracted from the Southampton City Council website or provided by City Council staff.

## Development of a policy appraisal framework

Three researchers (CV, DoC and DP-N) independently identified components of the WHO Action Plan relevant to maternal and child health, and dietary or physical activity behaviours that could be categorised into priority areas for the appraisal framework. Each researcher then highlighted policy options from the WHO Action Plan that were relevant for maternal (preconception, pregnancy, lactation and motherhood) and/or infant and child health. The policy options were specific initiatives that could be identified in policies. All policy options were independent from each other and corresponded with a single priority area. Supplementary appendix 1 presents the final appraisal framework agreed on by the three researchers. Discrepancies were resolved by returning to the WHO Action Plan to ensure consistency and accuracy.

A total of six priority areas were established for the appraisal framework. Three priority areas were action oriented: (1) maternal health, (2) infant/child health and (3) strengthening health systems. The remaining three represent public health processes necessary for high-quality policy development: (4) evidence-based strategies, (5) multisectoral action and (6) governance and accountability. Each priority area had between two and seven accompanying policy options (see additional material, supplementary appendix 1 for details). A total of 20 policy options were included in the national policy appraisal framework. The same 20 were included in the local policy appraisal framework because policy options such as fiscal, labelling, marketing or reformulation initiatives could also be employed in local settings such as schools, leisure centres or hospitals through local government commissioning contracts or guidelines. Three additional urban planning policy options were also included to encompass local government jurisdiction over urban planning. To ensure consistency in identifying the presence or absence of a policy option, search terms and inclusion and exclusion criteria for each policy option were created (additional material, supplementary appendix 1).

The policy options were categorised into three levels in line with the ecological model of health: individual, organisational and environmental (additional material, supplementary appendix 1). Seven policy options could not be categorised according to the three levels of the ecological model because they were not action oriented but represented good public health processes.

## Policy appraisal process

One researcher (DP-N) assessed all identified policies against the appraisal framework. Policies were reviewed individually and search terms (additional material, supplementary appendix 1) were used to pinpoint relevant sections that were comprehensively assessed to determine whether the policy option was present. The inclusion and exclusion criteria for each policy option were used to guide this procedure. Random selections, comprising 44% of the national policies and 50% of the

local policies, were double coded by a second researcher (SaS). Inconsistencies in coding were found in 4% of the double-coded policies, indicating over 95% level of agreement. Discrepancies were discussed with a third researcher (CV), and agreement was reached after returning to the policy and appraisal framework.

## Statistical analysis

Descriptive analyses were used to describe the policies by type, author and date of publication. The frequency for each of the policy options was calculated to ascertain their representation across the national and the local policies. Frequencies of policy options were also calculated according to the three national policy types: acts and codes, white papers and evidence-based guidelines. Cochran's Q test was used to test for differences in policy option frequency across the three ecological levels (individual, organisational and environmental) to identify whether there was difference in policy option adoption according to level of action. All statistical analyses were conducted using Stata statistical software package V.13.[22]

## Patient and public involvement

The public was not involved in this study, because it was a policy document analysis that did not involve participants.

## RESULTS
### Characteristics of identified policies

A total of 57 national policies relevant to maternal and child health were identified; 47 were published between May 2010 and June 2017 and 10 prior to 2010. Figure 1 provides a schematic representation of the national policy identification process.

Evidence-based guidance documents (n=27, 47%) and white papers (n=22, 39%) represented the vast majority of identified policies, with acts and codes representing the remainder (n=8, 14%). Additional material supplementary appendix 2 details the title, publication date and authors of each identified policy according to the three policy types. Of the 28 organisations that authored the national policies, the Department of Health was most prolific, authoring more than one-third of all policies (n=21, 37%; table 1), including half of the acts (n=4, 50%) and almost three quarters of the white papers (n=16, 73%). NICE was the second most common author (n=14, 25%), publishing the majority of the evidence-based guidance documents (n=14, 52%). Most policies were authored by a single organisation (n=46, 81%). However, 14 organisations were coauthors. The Department of Health was the most common coauthor (8 of the 11 coauthored policies).

Ten local policies were identified for appraisal, published between March 2011 and June 2017. The policies were authored by three different organisations (additional material supplementary appendix 2). The Southampton City Council led, or was involved in, the development of all local policies. Southampton City

**Table 1** List of authors for identified national policies

| Author | Number of single authored documents | Number of coauthored documents | Total* n (%) |
|---|---|---|---|
| Department of Health | 13 | 8 | 21 (37) |
| National Institute for Health Care Excellence | 14 | 0 | 14 (25) |
| Public Health England | 5 | 3 | 8 (14) |
| Department of Education† | 0 | 6 | 6 (11) |
| Local Government Association | 2 | 2 | 4 (7) |
| Advertising Standards Authority/Ofcom | 2 | 0 | 2 (4) |
| Department for Communities and Local Government | 1 | 1 | 2 (4) |
| Department for the Environment, Food and Rural Affairs | 2 | 0 | 2 (4) |
| Food Standards Agency | 1 | 1 | 2 (4) |
| Ministry of Justice | 1 | 1 | 2 (4) |
| National Health Service (NHS) England | 1 | 1 | 2 (4) |
| British Retail Consortium | 0 | 1 | 1 (2) |
| Cabinet Office | 0 | 1 | 1 (2) |
| Department for Business, Innovation and Skills‡ | 0 | 1 | 1 (2) |
| Department for Digital, Culture, Media and Sport | 0 | 1 | 1 (2) |
| Department for Work and Pensions | 0 | 1 | 1 (2) |
| Department of Transport | 1 | 0 | 1 (2) |
| Food Standards Scotland | 0 | 1 | 1 (2) |
| HM Revenue and Customs | 0 | 1 | 1 (2) |
| HM Treasury | 0 | 1 | 1 (2) |
| House of Commons Health Committee | 1 | 0 | 1 (2) |
| Mayor of London | 0 | 1 | 1 (2) |
| National Audit Office | 1 | 0 | 1 (2) |
| NHS Health Education England | 0 | 1 | 1 (2) |
| Northern Ireland Government | 0 | 1 | 1 (2) |
| Scientific Advisory Committee on Nutrition | 1 | 0 | 1 (2) |
| Town and Country Planning Association | 0 | 1 | 1 (2) |
| Welsh Government | 0 | 1 | 1 (2) |

*Column total reflects the total number of national policy documents published by each author and rounded percentage of all national policies (n=57).
†Department for Children, Schools and Families superseded by Department of Education.
‡Department for Business, Innovation and Skills superseded by Department for Business, Energy and Industrial Strategy,

Clinical Commissioning Group coauthored three policies (30%) and Southampton Safe City Partnership authored one policy (10%).

### Policy appraisal findings: national

Figure 2 shows the frequency of the priority areas and policy options across the 57 national policies. The priority areas that support public health processes were frequently included. The evidence-based practice and strengthening

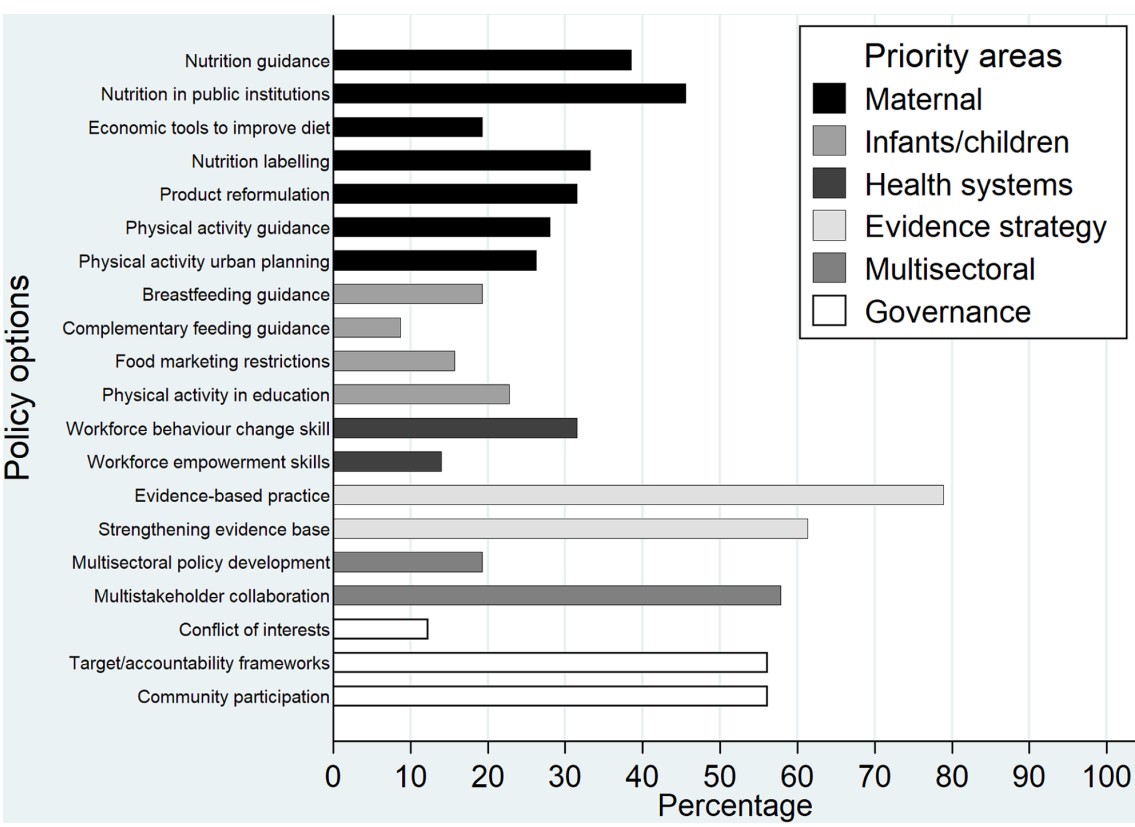

**Figure 2** Percentage of all 57 national policy documents for which each policy option in the policy appraisal framework was present.

the evidence base through research and evaluation policy options featured in 79% (n=45) and 61% (n=35) of the national policies, respectively. Community participation and accountability frameworks both featured in 56% (n=32) of the national policies. However, managing conflicts of interest was the second least frequent policy option (n=7, 12%). Multistakeholder collaboration was mentioned in more than half (n=33, 58%) of the national policies but multisectoral policy development was less common (n=11, 19%).

The action-oriented priority areas were less frequently included than those supporting good public health processes, with child health having the lowest prevalence (figure 2). Fewer than half (n=27, 47%) of the policies included one of the four infants/children policy options. Complementary feeding initiatives was the least common policy option, appearing in only 9% (n=5) of national policies, while children's food marketing restrictions featured in 16% (n=9). More than three quarters (n=46, 81%) of the policies contained one of the seven maternal policy options. Maternal nutrition related policy options (n=11, 19% to n=26, 46%) were better represented than those relating to maternal physical activity (n=15, 26% to n=16, 28%). One-third (n=20, 35%) of policies contained strategies to strengthen health systems. The need for workforce development in behaviour change strategies was cited in 32% (n=18) of policies, but workforce development in empowerment approaches was rarely mentioned (n=8, 14%).

### Policy content according to policy type

The frequencies of each policy option for the three national policy types are presented in figure 3. Among the eight acts and codes, nutrition labelling was the most common policy option appearing in 50% (n=4). Restrictions on marketing foods and beverages to children and evidence-based practice featured in 38% (n=3), while product reformulation, managing conflicts of interest and multisectoral collaboration were mentioned in 25% (n=2). Six policy options did not feature in any of the acts or codes. The vast majority of the 22 white papers included evidence-based practice and community participation (both n=18, 82%) and 73% (n=16) featured accountability frameworks and multisectoral collaboration. While nutrition in public institutions was the most common action-oriented policy option (n=11, 50%), developing workforce empowerment skills was not cited in any white papers and complementary feeding guidance was mentioned in only one (5%). Most of the 27 evidence-based guidance documents included evidence-based practices (n=24, 89%) and strengthening the evidence base through research (n=20, 74%). Nutrition guidelines featured in almost two-thirds (n=17, 63%) but managing conflicts of interest was mentioned in only one (4%).

### Policy content according to ecological model of health

Policy options to improve the macroenvironment were featured in half of the national policies (n=33, 58%).

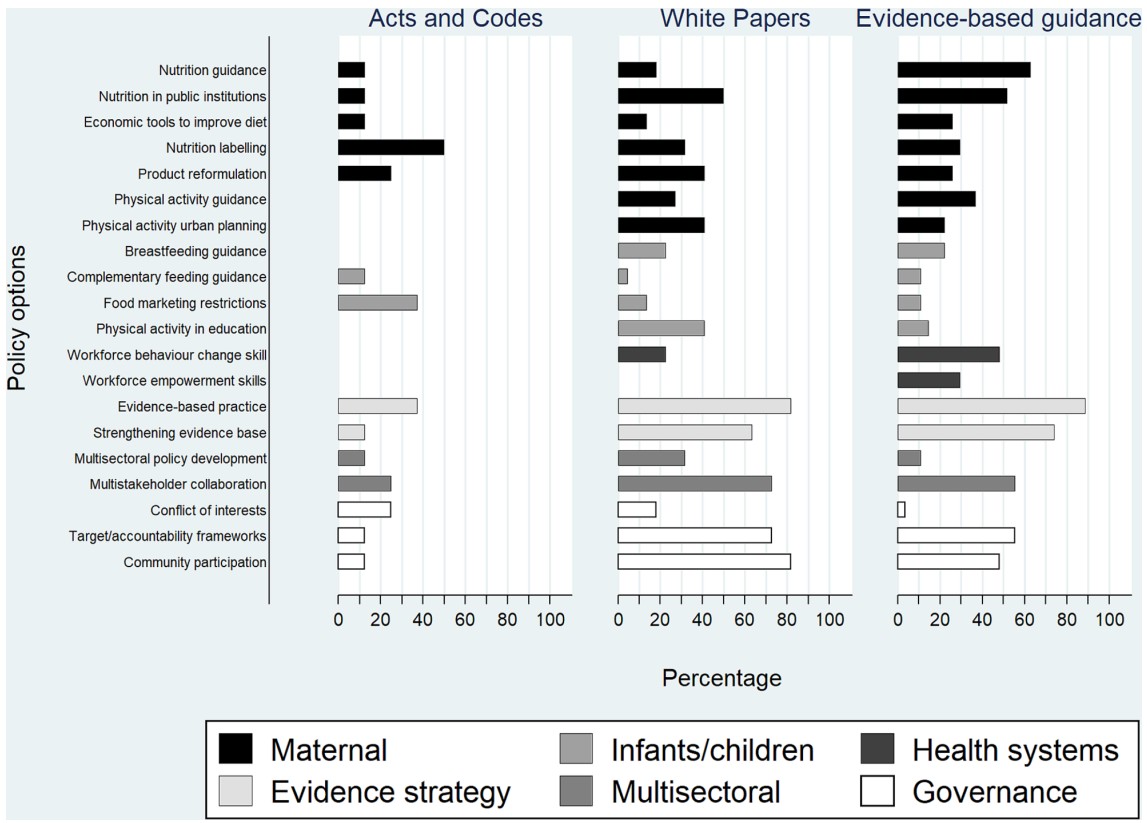

**Figure 3** Percentage of the 8 acts and codes, 22 white papers and 27 evidence-based guidance documents for which each policy option in the policy appraisal framework was present.

Policy options to create supportive social and physical environments in organisations featured more frequently (n=37, 65%), while those targeting individual level behaviour change were most common (n=41, 72%). Cochran's Q test revealed that there were no statistically significant differences (p=0.1) in representation of the ecological levels across the national policies.

**Policy appraisal findings: local**

Figure 4 shows the frequency of each policy option (including the three additional urban planning policy options) across the 10 local policies. Most of the local policies cited evidence-based practice (n=8, 80%), multisectoral collaboration (n=7, 70%) and multisectoral policy development (n=5, 50%). Maternal physical activity-related policy options had a strong representation with physical activity guidance, urban planning, improving green spaces and physical activity facilities and improving active transport infrastructure all appearing in 50% (n=5) of the local policies. Nutrition-focused policy options featured less frequently with five maternal nutrition options featuring in 10% (n=1) or less of the local policies. The two workforce development policy options also featured in only 10% (n=1) of local policies. Analysis against the ecological model of health revealed that macroenvironmental policy options featured in 90% (n=9) of the local policies, those targeting individual level behaviour change appeared in 50% (n=5) and those supporting healthy organisational settings in 30% (n=3).

Cochran Q test revealed that there was a statistically significant difference in representation of these ecological levels across the local policies (p=0.02).

**DISCUSSION**
**Summary of results**
Among the 57 national and 10 local policies identified in this study, policy options supporting public health processes were adopted more frequently than the action-oriented options targeting maternal and child dietary and physical activity behaviours. This finding was consistent across the three types of national policies and the local policies, although some differences were apparent. Multisectoral policy development was more common among the local than the national policies, while using targets to monitor policy implementation was more frequently included in the national policies. Managing conflicts of interest was rarely considered in the white papers or evidence-based guidelines but was addressed more often in the acts and codes, and local policies.

For the action-orientated policy options, those with a nutrition focus were more frequent than the physical activity ones among the national policies. Acts and codes in particular did not contain any physical activity policy options. The opposite was observed for local policies where there was a heavy focus on urban planning strategies to promote physical activity. Complementary feeding

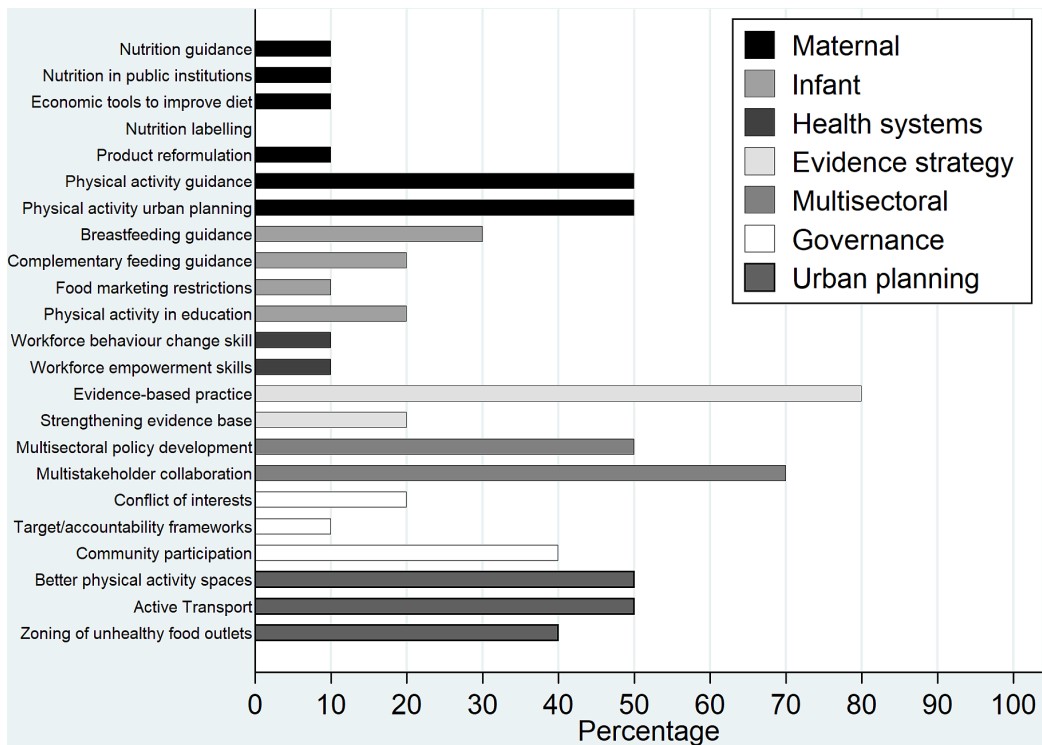

**Figure 4** Percentage of the 10 local policy documents for which each policy option in the policy appraisal framework was present.

guidance was virtually missing from all types of national policies, while nutrition labelling did not appear in any local policies. The need for workforce development in behaviour change skills was recognised across the white papers, evidence-based guideline and local policies, but workforce training in empowerment or self-care approaches was only cited in evidence-based documents and one local policy.

### Comparison with literature

This policy review is unique, compared with previous assessments of government action on obesity and NCDs in its focus on policy options specific to maternal, infant and child health. Existing policy frameworks consistent with the WHO Action Plan consider the entire population and measure progress for a specific behaviour or health outcomes. For example, the World Cancer Research Fund International NOURISHING framework and the International Network for Food and Obesity Research, Monitoring and Action Support Healthy Food Environment Policy Index (Food-Epi) considered policy activities for healthy eating and food environments,[23] while the WHO Global Monitoring Framework focuses on the prevalence of NCD risk factors.[17] Policy options that support the adoption of healthy nutrition and physical activity behaviours prior to conception (including adolescence) and during pregnancy, infancy and childhood have been recognised by the UK Chief Medical Officer as critical to prevent NCDs in future generations and help curb healthcare costs.[19]

The results of this study identified that, nationally, policy options related to maternal health were more frequently adopted than those related to child and infant health. General nutrition guidance and nutrition labelling were among the most frequent action-oriented policy options across the three types of national policies. Research from Thailand and England that used the Food-Epi framework similarly found good implementation of dietary guidelines and nutrition labelling when compared with international examples of good practice.[24 25] However, our study additionally assessed the inclusion of complementary feeding guidelines that we found to be the least well-adopted policy option. Complementary feeding is important for child development, growth and health, and this period of life presents a window of opportunity to prevent malnutrition in all forms.[26] Over the past few decades, progress on the inclusion of policy options to promote breastfeeding has been made, but this has not been matched by efforts to support complementary feeding, as shown by our study's results. Other countries have also shown limited success in meeting WHO complementary feeding recommendations[27 28] indicating that government action on this issue is needed.

A systematic review of the determinants of poor nutrition in children aged below 3 years identified that effective intervention strategies include improving maternal self-efficacy, being of sufficient intensity and reaching mothers during pregnancy.[29] Building the capacity of health professionals through workforce development is thus essential to engagement and supporting behaviour

change.[30] A review of policy options to address obesity in Spain identified workforce development in obesity prevention as one of the most popular approaches for governments to adopt.[31] The results of our policy review showed that workforce development in behaviour change skills was included in national English policies. However, recognition of the need to upskill healthcare professionals in empowerment approaches that improve self-efficacy was not included in any white papers, acts or codes. There is growing evidence that information is not enough to prompt behaviour change, particularly among disadvantaged groups, and that media campaigns increase inequalities in health behaviours.[32–34] While a number of the evidence-based documents reviewed in this study did include workforce empowerment approaches, the lack of coverage in white papers indicates that evidence is not adequate for policy option adoption. Policy decision making results from a combination of factors including the framing of an issue, political ideology, evidence of effectiveness and cost-effectiveness, timing and 'groundswell' or societal attitudes.[9 13]

Recent reviews of obesity-related policy analyses identified an overwhelming focus on national or state/provincial governments and a lack of consideration for how policy option adoption varies between national and local governments.[13 18] While limited to only one local authority, our study identified a clear difference in the focus of national and local policies. National documents contained a greater number of nutrition-related policy options, while local documents covered more policy options for physical activity, particularly macroenvironmental options. The transition of public health into local authorities in the UK has provided increased opportunities for health considerations in planning and environment policies.[35] The dominance of physical activity over dietary policy options in Southampton mirrors attitudes across other local councils in England. A consultation with staff from 14 councils identified low awareness of policy options for creating healthy food environments but good knowledge of planning activities to improve cycling and walking, and access to open spaces.[36] These findings suggest that the conditions for policy action, including public and political value, resources and operational capabilities[37] are each being met for physical activity strategies but not yet for population-level nutrition strategies.

There is a call for governments to enact regulation to create healthy food and physical activity environments as these policy options are consistently effective at improving lifestyle choices and do not widen inequalities in the way that information strategies can.[16] Our study showed that policy options to improve the macroenvironment were less well adopted by the English government than those targeting organisational settings or individual level behaviour change, although this difference was not statistically significant. Reviews that assessed international actions to address obesity across low-income, middle-income and high-income countries similarly identified

limited appetite from national governments for regulatory action and a preference for educational strategies.[38 39] The introduction of a UK sugar-sweetened beverage levy in 2018 indicates a growing global trend, led by action from governments in Mexico, France and Denmark, for stronger government actions to address obesity. However, concerns about adequate enforcement and the effectiveness of self-regulatory and voluntary codes, such as those used to govern restrictions for online advertising of unhealthy food to children or the reformulation of sugar and salt in foods, indicate that tougher regulatory action in these areas is still needed.[38]

Similar to reviews of the implementation of national policies related to food environments in New Zealand, Thailand and England,[24 25 40] we found that policy options supporting sound public health processes were frequently included across policy types. However, this study additionally identified that only a small proportion of English policies contained details about managing conflicts of interest. This issue is important for governments to consider during policy development and adoption because strategies to improve maternal and child dietary and physical activity behaviours could impact on a number of profit-driven commercial industries. In particular, commercial interests from infant formula manufacturing companies directly compete with breast feeding.[41] Strategies to manage conflicts of interest were not included in a number of policies where potential conflicts are likely including The Processed Cereal-based Foods and Baby Foods for Infants and Young Children (England) Regulations. Limiting commercial influence and developing clear processes for accountability of both the public and the private sector are necessary.[23]

### Implications for policy and research

The UK Government's current childhood obesity strategy outlined few measures for early childhood despite the WHO's Commission on Ending Childhood Obesity recommending member states develop clear complementary feeding guidance, including portion sizes and recommendations to avoid sugar-sweetened beverages and energy-dense nutrient-poor foods. The much-awaited review by the Scientific Advisory Committee on Nutrition on complementary feeding will create an opportunity for the publication of clear guidelines in national white papers and their adoption by local and national agencies.

Health and social care services that support behaviour change by empowering individuals to generate their own solutions have shown beneficial effects on the self-efficacy of women from disadvantaged populations.[42] Practitioners from a range of backgrounds can be successfully trained in empowerment skills and these offer great potential to be used in routine health and social care to support behaviour change among harder to reach women and families.[43] The NHS 'making every contact count' movement recognises these opportunities for contact, but further leadership is required from national government to include this policy option in white papers and

systematically build workforce capacity in empowerment skills across England.

Leadership and funding from state and national governments have been shown to affect policy priorities at the local level. Local governments in the State of Victoria in Australia have higher adoption of policy initiatives to support physical activity and healthy eating than other states. This increased activity has been directly attributed to the mandatory requirement under the Victorian Health Act 1958 and the Planning and Environment Act 1987 for local governments to prepare Municipal Public Health Plans, in addition to the offer of financial incentives.[9 44] In the UK, The National Planning Policy Framework requires planning departments to promote healthy communities, taking into account and supporting local Joint Health and Wellbeing Strategies. This national leadership has shown to be beneficial for physical activity-related policy options, but local public health teams now need to work strategically with planning and environmental health colleagues to support more healthy eating opportunities including zoning restrictions of takeaways or licencing requirements for the sale of healthy foods.[36] Collaborations between local universities and public health teams could provide necessary evidence for local government action to support development of healthy food environments, as local evidence provides impetus for local government action.[9 45] Similarly, supranational unions, such as the European Union (EU), have provided leadership or hindrance for member states directing initiatives related to the analysis conducted in this study. While it was beyond the scope of this study to assess EU governance over English policies, Brexit provides opportunities for brave policy changes that are supportive of healthy dietary and physical activity behaviours.

Recent evidence from a leaked email shows that Coca-Cola placed indirect pressure on governments through an industry-funded scientific institute to influence policy to benefit commercial companies rather than the public.[46] Public confidence can be damaged if government's management of conflicts of interest is not sufficiently documented.[47] By using the international best practice standards for managing conflicts of interest, governments in England could appropriately manage this issue going forward.[48]

### Strengths and limitations

This study is one of few policy reviews to focus on maternal, infant and child health for NCD prevention and to consider government activity at both the national and local levels. The study involved a comprehensive desktop search for three types of policies that was aided by senior health officials in local and national governments to enhance completeness of the search strategy. Strengths of the appraisal framework include that it mirrored current international best practice standards, was developed and was tested by three researchers independently. For quality assurance, one author completed the data extraction for all identified policies and half were also coded by a second researcher.

This study also had a number of limitations. Some policies may have been missed because they were not available electronically or were not identified by our search. The local policy results for Southampton may not be representative of action in other local authorities. In addition, the appraisal framework and data extraction results were based on secondary data analysis only. They were not validated through extensive consultation with policy makers and other expert stakeholders due to resource limitations. However, key search terms and inclusion and exclusion criteria for each policy option were used to increase accuracy of the document assessment. Furthermore, the use of a dichotomous measure does not provide comprehensive details about the scale of policy option implementation, and not all policy options would be expected to appear in every policy. However, adopting a quantitative approach enabled assessment of overall adherence to WHO international best practice and compared the level of activity between priority areas allowing easy identification of areas for improvement. This approach could be used by governments to monitor progress on NCD prevention targeting women and children.

### CONCLUSION

This study assessed policies relevant to improving maternal and child dietary and physical activity behaviours against the WHO international standard. It aids understanding of the national and local progress in England of adopting policy options to prevent NCDs. Analyses of 57 national policies highlighted greater inclusion of policy options related to maternal than child health (particularly complementary feeding initiatives) and individually focused policy options (such as nutrition labelling) than environmental strategies. Action at the local level showed a stronger focus on policy options for physical activity than those for healthy eating and for child health than for maternal health. Policy options that underpin good public health processes such as evidence-based strategies and multisectoral collaboration were widely adopted by both national and local governments, but further action is needed by policy makers to manage conflicts of interest and curb industry lobbying during policy development. Opportunities exist for the research community and health professionals to influence policy in a meaningful way. Such opportunities include forming coalitions to mobilise societal support and framing issues as solutions in a way that is palatable to policy makers. The timing of external events can positively influence policy decision making. The 2012 Olympics provided impetus for policy action supportive of physical activity.[49] Challenging as the management of Brexit is, it provides a unique opportunity for policy change that holds great potential to reform food and transport systems to be supportive of healthy dietary and physical activity behaviours.

**Author affiliations**
[1]Medical Research Council Lifecourse Epidemiology Unit, University of Southampton, Southampton, UK
[2]Liggins Institute, University of Auckland, Auckland, New Zealand
[3]NIHR Southampton Biomedical Research Centre, University Hospital Southampton NHS Foundation Trust and University of Southampton, Southampton, UK
[4]Public Health, Southampton City Council, Southampton, UK

**Acknowledgements** We are grateful to Patsy Coakley for computing support and to the programme managers at Public Health England for their support with policy identification. We would like to thank our funders for their support.

**Contributors** DP-N and CV conceived the study and wrote the first draft of the manuscript. HI, JB, MB, DeC, CC and WL contributed to the study design and helped draft the manuscript. DP-N, TR, SoS and DeC identified the relevant policies. DP-N, DoC and CV designed the policy audit tool. DP-N, SaS, DoC and CV conducted the policy audit. All authors read and approved of the manuscript.

**Funding** This work was supported by funding from: NIHR Southampton Biomedical Research Centre, University of Southampton University Hospital Southampton NHS Foundation Trust. European Union's Seventh Framework Programme (FP7/2007-2013), project EarlyNutrition (289346). CV was supported by a fellowship from the University of Southampton (PCTA36/2015).

**Disclaimer** The views expressed in this publication are those of the authors and not necessarily those of the funding organisations.

**Competing interests** DP-N, SaS, DoC, DeC, HI and CV have no conflicts of interest to declare. SoS, TR, WL, MB and JB have received grant research support from Danone Nutricia Early Life Nutrition. CC has has received consultancy, lecture fees and honoraria from AMGEN, GKS, Alliance for Better Bone Health, MSD, Eli Lilly, Pfizer, Novartis, Servier, Medtronic and Roche. The study described in this manuscript is not related to these conflicted relationships.

**Patient consent** Not required.

**Provenance and peer review** Not commissioned; externally peer reviewed.

**Data sharing statement** The data are available upon request to the corresponding author.

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
