## [Reviewer comments · BMJ Open]

ARTICLE DETAILS

TITLE (PROVISIONAL)	How well do national and local policies in England relevant to maternal and child health meet the international standard for non-communicable disease prevention? A policy analysis
AUTHORS	Penn-Newman, Daniel; Shaw, Sarah; Congalton, Donna; Strommer, Sofia; Rose, Taylor; Lawrence, Wendy; Chase, Debbie; Cooper, Cyrus; Barker, Mary; Baird, Janis; Inskip, Hazel; Vogel, C

VERSION 1 – REVIEW

REVIEWER	Ana Raquel Matos Centre for Social Studies, University of Coimbra – Portugal
REVIEW RETURNED	21-Mar-2018

GENERAL COMMENTS	Authors should pay attention to the following questions: a) there are long sentences that make the reading more difficult, for example, the sentence of the abstract - objective; b) there are some typing mistakes, including in the final references; c) although authors have opted for a quantitative approach to the collected data, they neglect a whole qualitative approach that sustains the article. In fact, the article presented is based on secondary data analysis, namely documentary analysis, which is not even mentioned in the methodology. This safeguard should at least be made clear during the study limitations discussion.
---

REVIEWER	Anne Marie Thow University of Sydney, Australia
REVIEW RETURNED	20-Apr-2018

GENERAL COMMENTS	This study addresses an important issue for public health policy. The paper presents an analysis of maternal child health policy, with respect to NCD prevention and control in the UK at national level and at local government level (Southampton City). The authors have reviewed an impressive number of policy documents, but I found it difficult to discern the meaning of the findings presented for assessing implementation of NCD policy in the UK. My main concerns regarding the analysis presented are:  1. Key terms are not defined. In particular, there is no definition of Maternal Child Health policy, which makes it difficult to assess the conceptual basis of the study (I mean, given the list of policies I would expect discussion of the rationale for including policies that impact on maternal child health but do not explicitly articulate this. For example, education policies, social policies). Neither are search terms are provided in the methods. 2. Similarly, the term 'implementation' is used throughout the
--

	document as the focus of the policy analysis – but the term is not defined and it appears to be used as a proxy for policy adoption (not implementation) or even just to refer to an issue being mentioned in a policy document. In relation to this point, the analysis presented makes no distinction between an issue being raised in a policy document (for example, in a normative sense, such as fiscal policy being a useful tool in this context) and a policy directive being issued (e.g. the UK soft drinks levy). So interpreting statements such as ‘economic tools and food marketing restrictions were featured in half of the national policies (58%)’ (p12, line 46-48) is basically impossible. At the very least, this distinction needs to be made and the findings clarified with reference to this. 3. I was also unclear of the meaning of ‘evidence-based practice’ as a category for policy content (throughout, e.g. p12, line 24-25). Surely the area of practice is important to specify, as it would make a difference in terms of where it would be appropriate for the issue to be addressed? 4. The value of quantitatively assessing the presence of policy options is not clear to me. There seems to be a general lack of reflexivity regarding the normative assumptions made (or not explored) in the analysis. For example, ‘more than three quarters of policies contained one of the seven maternal policy options’ (p11, line 48). Why is it meaningful to group these policies together for quantitative presentation? Are they mutually exclusive, such that the presence of any one option is interchangeable? Would it matter if what this actually meant was that one policy option was present a number of times, which others were completely absent? 5. Generally, I found myself confused by the meaningfulness of the focus on how many times a policy option was mentioned, without any indication of where or how it should be mentioned, in order to generate action and outcomes in the right place. One of the best definitions of policy that I have encountered is that of the UK Cabinet Office, which defined policy as translating a ‘vision into programmes and actions to deliver ‘outcomes’ – desired changes in the real world’ (dera.ioe.ac.uk/6320/1/profpolicymaking.pdf). The analysis presented in this paper is missing any consideration of 1) jurisdictional responsibilities (which a comparison of national and local policy should entail) and 2) whether the points at which it was mentioned were sufficient to generate meaningful change in the real world. For example, on p 12 lines 3-5 the authors state ‘workforce development in empowerment approaches was rarely mentioned (14%)’, without any reference to the context in which it was mentioned, and why this might be insufficient for action in the real world, and on p14 line 21, the authors state ‘complementary feeding guidance was virtually missing from all type of national policies, while nutrition labelling did not appear in any local policies’, without any consideration of whether jurisdictional responsibilities might have precluded this and whether (again) where it was addressed was (or wasn’t) sufficient to generate the desired outcomes. 6. The cross sectoral nature of NCD policy is not considered in any meaningful detail. E.g. the statement on p9, line 53 that ‘14 organizations were only co-authors’ (emphasis added). Why ‘only’? Isn’t cross sectoral policy making a good thing for obesity prevention? 7. The metrics presented in the abstract findings are difficult to interpret and do not align with the appraisal framework as described on p7, lines 40-46
--	--

REVIEWER	Chris van Weel Radboud University Nijmegen, Department of Primary and Community Care, The Netherlands
REVIEW RETURNED	24-May-2018

GENERAL COMMENTS	This paper analyses the extent to which national and local policies for maternal and child health meet available international standards. The findings are that there are gaps and that on a national level food and nutrition practice are particularly developed, while on the local level there is an emphasis on physical activity. The paper is well written and the data have been collected in a clear and robust way. My main point with the paper is its lack of critical reflection on the role of public health, in relation to available evidence. It looks a bit like 'when X guidelines are available, X should also be implemented'. I would argue that public health campaigns could benefit as much from 'focus' rather than completeness. Therefore I would ask the authors to address this:  1. What is the current UK public health priority and focus, and how many of the international guidelines etc. available, were in line with this priority? 2. How many guidelines and policy documents were considered, when presenting %? It is not always clear in the text on what denominator presented % were based. (See figures 2, 3 4). And how complete was their search? 3. The authors describe the various documents, but I can find little information on their actual content, that may vary from general recommendations to detailed guidance for practice. The authors looked at local versus national policy, but this would benefit from more detail:  4. There must be a pre-determined task division in public health of what would suit a local and what a national policy. The fact that nutrition is more adhered to in national policy while physical activity finds more often its way to local policy could be a surprise finding or reflect a predetermined task division between the two levels. I can imagine that this relates to issues like taxation and product-safety requirements (a national jurisdiction) and planning roads (a local authorities jurisdiction). 5. How did the authors deal with local needs – as local communities may have needs differing from the global perspective the authors had drawn their guidelines from? 6. The paper does not refer in any way to the interface between public health and primary care – a strength of the UK health system: to what extent were guidelines transferred into UK primary health care rather than pursued through local policy? 7. On page 9, lines 46 – 55 findings very much reflect a top-down approach to policy setting and implementation. However, to over needs of local populations a bottom-up approach. It is not clear how this bottom-up part of the policy setting has been considered in this analysis. The nature of the various documents may require a bit more detail: it is great to have a WHO endorsed document. But WHO documents are often based on model documents from its member countries, and in that way there is the possibility of no more than double-up existing UK policy documents. That way, it is not so much 'international guidance' but existing UK policy.  8. How often did the authors find such overlap in their search?
--

	9. Like this reviewer, readers will need a bit more details on the various categories the authors constructed. For example, what were the non-action oriented policy options (page8, lines 14 – 21? 10. The same is true for the findings presented on page 9: what were the policies, guidelines etc found? Keep in mind the international readership of the BMJ Open. 11. I am puzzled with what the authors mean by 'conflict of interest' acts and codes, in the context of this analysis. Probably giving examples could clarify this. 12. Could the authors expand a bit on their statement on page 16, lines 16 – 18 that evidence alone is not sufficient for policy implementation? Where are the limits of evidence and what 'best practice' is available to back this up? And to what extent is this an international best practice, transferrable to other settings, to what extent is this politics-dependent national or local business? My final point is that in essence this is all about process – interventions to be implemented. The real interest is in its final outcome: 13. The analysis as I stated under point 1 looks very much a search on whether all that is available is implemented. But with what avail: which guidelines, policy documents etc were just a better way of doing what already is quite well pursued in public health, and to what extent does it hold the promise of what has until now not been very successfully launched?
--	---

VERSION 1 – AUTHOR RESPONSE

Reviewer 1

1. There are long sentences that make the reading more difficult, for example, the sentence of the abstract objective.

We thank the reviewer for this comment and have amended the manuscript to shorten sentences or objectives where they were unnecessarily long. See below the abstract objective. See the revised manuscript for changes on pages, 5, 6, 8, 16, 17, 21 and 22.

Abstract

Objectives: i) To identify national policies for England and local policies for Southampton City that are relevant to maternal and child health. ii) To quantify the extent to which these policies meet the international standards for nutrition and physical activity initiatives set out in the World Health Organisation Global Action Plan for the Prevention and Control of Non-Communicable Diseases (NCDs) (WHO Action Plan).

2. There are some typing mistakes, including in the references

Thank you for highlighting these errors. We have now made corrections throughout the manuscript and to the reference list.

3. Although authors have opted for a quantitative approach to the collected data, they neglect a whole qualitative approach that sustains the article. In fact, the article presented is based on secondary data analysis, namely documentary analysis, which is not even mentioned in the methodology. This safeguard should at least be made clear during the study limitations discussion.

We acknowledge this methodological consideration raised by the reviewer. We have amended a sentence in the methods section outlining the secondary analysis nature of this study (page 6). We have also added a sentence in the limitations section of the discussion (page 21).

Page 6

Ethical approval was not required for this study because it was a secondary analysis of publicly available policy documents.

Page 21

In addition, the appraisal framework and data extraction results were based on secondary data analysis only. They were not validated through consultation with policy makers and other expert stakeholders due to resource limitations. However, key search terms and inclusion and exclusion criteria for each policy option were used to increase accuracy of the document assessment.

Reviewer 2:

This study addresses an important issue for public health policy. The paper presents an analysis of maternal child health policy, with respect to NCD prevention and control in the UK at national level and at a local level (Southampton City). The authors have reviewed an impressive number of policy documents, but I found it difficult to discern the meaning of the findings presented for assessing implementation of NCD policy in the UK.

1. Key terms are not defined. In particular, there is no definition of maternal and child health policy, which makes it difficult to assess the conceptual basis of the study (I mean given the list of policies I would expect discussion of the rationale for including policies that impact on maternal child health but do not explicitly articulate this). For example, educational policies, social policies. Neither are search terms provided in the methods.

Thank you for highlighting the need for a definition of maternal and child health policies in the methods section. We have now included this definition in the methods section (page 7 and below). Our method did not involve using search terms to identify policies. Rather, exhaustive searches of the websites of government departments and agencies that can influence government policy development were completed. The search involved screening all documents to identify those relevant to maternal and child health or those that could influence the health or dietary or physical activity behaviours of women and children. The search was then verified by a programme manager at Public Health England. We have added further details to the methods section about this process (page 7). Search terms governing the policy appraisal have been provided in Additional material, Appendix 1 because they are specific to each policy option and are too numerous to mention in the text. Appendix 1 is cross-referenced in the text (page 8).

Page 7

At each time point, government websites were exhaustively searched to identify acts and codes, briefing notes, white papers and evidence-based guidelines that were relevant to maternal and child health. We adopted a broad definition of maternal and child health policy by including policy documents that targeted the health of women and/or children, as well as those that could indirectly influence the health or dietary or physical activity behaviours of women and children.

2. Similarly, the term 'implementation' is used throughout the document as the focus of the policy analysis – but the term is not defined and it appears to be used as a proxy for policy adoption (not implementation) or even just to refer to an issue being mentioned in a policy document. In relation to this point, the analysis presented makes no distinction between an issue being raised in a policy document (for example, in a normative sense, such as fiscal policy being a useful tool in this context) and a policy directive being issues (e.g. the UK soft drink levy). So interpreting statements such as

'economic tools and food marketing restrictions were featured in half of the national policies (58%)' (pg12, line 46-48) is basically impossible. At the very least, this distinction needs to be made and the findings clarified with reference to this.

We are grateful to the reviewer for this comment and, on reflection, agree that 'implementation' may not be the most appropriate term to describe the results of this study. Throughout the manuscript we have replaced 'implemented' with 'adopted' or 'included'. We have retained the sentence stating that this study does not provide comprehensive details about the scale of policy option implementation in the limitations section of the discussion (page 21).

With regards to the reviewer's second point, it is for this reason that we provided analyses of the policy options according to the three policy types. Acts and codes have different levels of authority to white papers and evidence-based guidance. We have added extra detail to the definitions of these types of policy documents in the introduction to clarify this issue to the reader (pages 4-5, and below). The specific sentence in the results section to which the reviewer refers above is in fact answering a different question. It collectively describes the difference in representation of policy options across three levels of an ecological model of health (environmental, organisational, individual) to identify if the levels are equally represented as recommended in health promotion theory. This sentence provides an answer to the third aim of the study and is accurately presented. We recognise the confusion that providing the policy option examples creates in this sentence and have therefore removed them (page 13). Complete details of the appraisal framework policy options relevant to each ecological level are provided in submitted Additional material, Appendix 1.

Page 4-5

National government policies cover a variety of document types that hold different levels of authority from clear government directive to recommending an action in a normative sense. One set of categories applicable in the UK includes: i) acts and codes - legally binding approved by the House of Commons and Lords that are implemented by an appropriate government agency [10], ii) white papers – an outline of government's strategic direction or priorities for action on a particular subject [11], and iii) evidence-based guidelines – implicit policies including summaries of evidence into advice on best-practice standards or recommended course of action, such as guidance by the National Institute for Health and Care Excellence (NICE).[12] Different types of national policies may vary in scope or represent comprehensive, complementary action depending on a government's political ideology, time in office and power across the parliamentary houses, amongst other factors.[13] Policy options to address an issue are often raised initially in policy documents with less authority, to be discussed and debated in parliament before adoption as a government directive. Assessing similarities across or differences between types of policy documents can provide insight into areas for improvement, particularly when compared to international standards.

3. I was also unclear of the meaning of 'evidence-based practice' as a category for policy content (throughout, e.g. p12, line 24-25). Surely the area of practice is important to specify, as it would make a difference in terms of where it would be appropriate for the issue to be addressed?

We thank the reviewer for the opportunity to clarify this term. The appraisal framework developed in this study includes two evidence-based policy options that were drawn from the overarching principles of the WHO Action Plan. The definitions of these two evidence-based policy options are provided in Additional Material Appendix 1 of the manuscript. The first policy option is evidence-based practice which must refer to the use of evidence-based strategies, particularly the latest scientific evidence and/or best practice guidelines that informed the development of policies, programmes or interventions. The second policy option is strengthening the evidence base for effective dietary and physical activity interventions through research and evaluation. We believe that these definitions provided clear guidance for data extraction whereby the researchers considered the evidence-base

for strategies included in the policy being reviewed with particular focus on those related to diet and physical activity. We have included another cross-reference to Appendix 1 in the methods section of the manuscript to highlight to the reader where these definitions are provided (page 8).

4. The value of quantitatively assessing the presence of the policy options is not clear to me. There seems to be a general lack of reflexivity regarding normative assumptions made (or not explored) in the analysis. For example, 'more than three quarters of policies contained one of the seven maternal policy options' (p11, line 48). Why is it meaningful to group these policies together for quantitative presentation? Are they mutually exclusive, such that the presence of any one option is interchangeable? Would it matter if what is actually meant was that one policy option was present a number of times, which others were completely absent?

We thank the reviewer for the opportunity to explain the value in our quantitative approach. The WHO Action Plan provides a suite of policy options that if implemented collectively will provide an international gold standard for action against non-communicable diseases (NCDs). By adopting a quantitative approach to policy analysis we were able to not only assess overall adherence to WHO best practice but also compare the level of activity between priority areas. These analyses provide information about areas for improvement. Using the example mentioned by the reviewer, our results identified that more than three quarters of policies contained at least one of the seven maternal policy options, however fewer than half included one of the four infant/child policy options. These findings indicate that greater attention could be given to adopting the recommended child policy options in the future to enhance action for prevention of NCDs relevant to dietary and physical activity behaviours. We are grateful to the reviewer for highlighting the need for us to address this issue early in the manuscript and we have included additional sentences in the introduction (page 6 and below) and discussion (page 21 and below).

Page 6

Assessment of how well national and local government policies are supporting women and children to adopt healthy lifestyles is needed to identify gaps for further government action. Therefore, this study aimed to quantify the extent to which English national and local government policies that are relevant to maternal and child health meet the international standard set out in the WHO Action Plan.

Page 21

However, adopting a quantitative approach enabled assessment of overall adherence to WHO international best practice and compared the level of activity between priority areas allowing easy identification of areas for improvement.

5. Generally, I found myself confused by the meaningfulness of the focus on how many times a policy option was mentioned, without any indication of where or how it should be mentioned, in order to generate action and outcomes in the right place. One of the best definitions of policy that I have encountered is that of the UK Cabinet Office, which defined policy as translating a 'vision into programmes and actions to deliver 'outcomes' – desired changes in the real world' (dera.ioe.ac.uk/6320/1/profpolicymaking.pdf). The analysis presented in this paper is missing any consideration of 1) jurisdictional responsibilities (which a comparison of national and local policy should entail) and 2) whether the points at which it was mentioned were sufficient to generate meaningful change in the real world. For example, on p 12 lines 3-5 the authors state 'workforce development in empowerment approaches was rarely mentioned (14%)', without any reference to the context in which it was mentioned, and why this might be insufficient for action in the real world, and on p14 line 21, the authors state 'complementary feeding guidance was virtually missing from all type of national policies, while nutrition labelling did not appear in any local policies', without any consideration of whether jurisdictional responsibilities might have precluded this and whether (again) where it was addressed was (or wasn't) sufficient to generate the desired outcomes.

As highlighted in the discussion of our manuscript, it is a limitation that this study does not provide comprehensive details about the scale of policy option implementation. However, as described above in point 4, the quantitative assessment does provide an indication of overall level of activity on a policy option and enables easy identification of policy options and priority areas that need further action. Additionally assessment according to policy type (acts, white papers, evidence-based guidance and local, national) provides an indication of where action is needed by different spheres of government. With regards to differences in jurisdictional responsibilities, we defined the policy options in a manner that was meaningful to both national and local governments to enable comparison. For example, while nutrition labelling may only be legislated at a national government level, we included labelling activities that are possible at a local or institutional level such as nutritional menu prompts or shelf prompts that can be implemented within schools, hospitals or leisure facilities where governments hold contracts with providers. We thank the reviewer for highlighting the fact that we had only made these definitions clear in the Additional material and have now included sentences outlining this in the introduction (page 4 and below) and methods sections (page 8 and below).

Page 4

While jurisdictional responsibilities differ between government levels and only national governments can enact legislation, for example on nutrition labelling, there are activities that local governments can adopt on the same issue, such as shelf or menu nutrition prompts in cafeterias in schools, hospitals or leisure centres.

Page 8

The same 20 were included in the local policy appraisal framework because policy options such as fiscal, labelling or marketing initiatives could also be employed in local settings such as schools, leisure centres or hospitals through local government commissioning contracts or guidelines.

6. The cross sectoral nature of NCD policy is not considered in any meaningful detail. E.g. the statement on p9, line 53 that '14 organizations were only co-authors' (emphasis added). Why 'only'? Isn't cross sectoral policy making a good thing for obesity prevention?

We thank the reviewer for highlighting that this sentence was confusing and we have removed 'only'. We feel that we covered multisectoral action in a meaningful manner in this study because multisectoral action was a separate priority area that included two policy options that were assessed in each of the 57 national and 10 local policies identified. The two policy options included multisectoral development and multistakeholder collaboration. The definitions of each are included in Additional material Appendix 1. To improve clarity we have reworded the description of the policy framework (page 8 and below).

Page 8

A total of six priority areas were established for the appraisal framework. Three priority areas were action-orientated: i) maternal health, ii) infant/child health, iii) strengthening health systems. The remaining three represent public health processes necessary for high-quality policy development: iv) evidence-based strategies, v) multisectoral action, and vi) governance and accountability. Each priority area had between two and seven accompanying policy options (see Additional material, Appendix 1 for details).

7. The metrics presented in the abstract findings are difficult to interpret and do not align with the appraisal framework as described on p7, lines 40-46

We have reworked the results section of the abstract to describe key findings for both the priority areas and the policy options as both are described in the appraisal framework.

Abstract

Results: Across both national and local policies, priority areas supporting public health processes, such as evidence-based practice, were adopted more frequently than the action-orientated areas targeting maternal and child dietary and physical activity behaviours. However, the policy option managing conflicts of interest was rarely considered in the national policies (12%), particularly in white papers or evidence-based guidelines. For the action-orientated priority areas, maternal health policy options were more frequently considered than those related to child health or strengthening health systems. Complementary feeding guidance (9%) and workforce training in empowerment skills (14%) were the least frequently action-orientated policy options adopted among the national policies. The maternal nutrition-focused and workforce development policy options were least frequent among local policies adopted in 10% or fewer. Macro-environmental policy options tended to have a lower priority than organisational or individual options among national policies ($p=0.1$) but had higher priority among local policies ($p=0.02$).

Reviewer 3:

This paper analyses the extent to which national and local policies for maternal and child health meet available international standards. The findings are that there are gaps and that on a national level food and nutrition practice are particularly developed, while on the local level there is an emphasis on physical activity. The paper is well written and the data have been collected in a clear and robust way.

We thank the reviewer for these comments.

1. What is the current UK public health priority and focus, and how many of the international guidelines etc. available, were in line with this priority?

We have added a sentence to the introduction (page 6 and below) outlining that non-communicable disease (NCD) prevention, particularly addressing the burden of obesity, are a key public health priority in the UK. Obesity is a persistent public health problem affecting many countries and the need to adopt policy options to address obesity has been recognised by international organisations such as the World Health Organisation (WHO) and the United Nations. The WHO Action Plan used in this study recognises the primary role of governments in tackling obesity and sets out a range of policy options which equate to the current international best practice standard.

Page 6

Additionally, the public health white paper for England 'Healthy Lives Healthy People' and the 'Childhood Obesity' strategy highlight the need for action across government sectors and levels to address the persistent public health issue of obesity. Assessment of how well national and local government policies are supporting women and children to adopt healthy dietary and physical activity behaviours is needed to identify gaps for further government action.

2. How many guidelines and policy documents were considered, when presenting %? It is not always clear in the text on what denominator presented % were based. (See figures 2, 3 4). And how complete was their search?

We thank the reviewer for their comment and have included the number of policies next to each percentage to improve clarity for the reader throughout the results section (pages 12-14). The denominator for each Figure is provided in the Figure title.

With reference to the completeness of the policy search, we conducted an exhaustive search of government department and executive agency websites and had the searches verified by a national

and a local policy maker to identify missing policy documents (pages 7-8 and below). We included a sentence in the limitations section of the discussion (page 21 and below).

Page 7

The compilation of the identified national policies was then sent to a programme manager at Public Health England for verification and identification of missing policies.

Page 8

To aid methodological rigor and completeness, the identification of local Southampton policies was supported and verified by a public health consultant (DChase) at the Southampton City Council.

Page 21

Some policies may have been missed because they were not available electronically or were not identified by our search.

3. The authors describe the various documents, but I can find little information on their actual content, that may vary from general recommendations to detailed guidance for practice.

We appreciate the point that is being raised by the reviewer. We have provided a table of title, authors and publication date of each of the 57 national and 10 local policies reviewed in Additional materials Appendix 2. The national policies were grouped according to policy type (acts and codes, white papers and evidence guidelines) and definitions of these policy types are provided in the introduction (page 4). On page 10 (and below) we direct the reader to this additional information about the 67 policies reviewed. Due to word count limits we are unable to provide more detailed descriptions about the content of the policy documents.

Page 10

Additional material Appendix 2 details the title, publication date and authors of each identified policy according to the three policy types.

4. There must be a pre-determined task division in public health of what would suit a local and what a national policy. The fact that nutrition is more adhered to in national policy while physical activity finds more often its way to local policy could be a surprise finding or reflect a predetermined task division between the two levels. I can imagine that this relates to issues like taxation and product-safety requirements (a national jurisdiction) and planning roads (a local authorities jurisdiction).

Differences in jurisdictional responsibilities were considered when we created definitions for each policy option. We were careful to ensure definitions were meaningful to both national and local governments to enable comparison. For example, while nutrition labelling may only be legislated at a national government level, we included labelling activities that are possible at a local or institutional level such as nutritional menu prompts or shelf prompts that can be implemented within schools, hospitals or leisure facilities where governments hold contracts with providers. We thank the reviewer for highlighting the fact that we had only made these definitions clear in the Additional material and have now included sentences outlining this in the introduction (page 4 and below) and methods sections (page 8 and below).

Page 4

While jurisdictional responsibilities differ between government levels and only national governments can enact legislation, for example nutrition labelling, there are activities that local governments can adopt on the same issue, such as shelf or menu nutrition prompts in cafeterias in schools, hospitals or leisure centres.

Page 8

The same 20 were included in the local policy appraisal framework because policy options such as fiscal, labelling or marketing initiatives could also be employed in local settings such as schools, leisure centres or hospitals through local government commissioning contracts or guidelines.

5. How did the authors deal with local needs – as local communities may have needs differing from the global perspective the authors had drawn their guidelines from?

We thank the reviewer for the opportunity to clarify this point. In the methods section of the manuscript, we described the prevalence of overweight and obesity among young women in the City of Southampton. We have amended this sentence on page 7 (and below) to make it clearer for the reader that this public health issue is similar to the national level and the international recognition of the need to address growing levels of obesity among women and children.

Page 7

In 2014/15, 47% of women in Southampton were classified as overweight or obese at their maternity booking appointment,[21] indicating that this local health issue is similar to the national and international level.

6. The paper does not refer in any way to the interface between public health and primary care – a strength of the UK health system: to what extent were guidelines transferred into UK primary health care rather than pursued through local policy?

We thank the reviewer for the opportunity to clarify this point. We acknowledged the interface between public health and primary care by including a broad range of policy documents from a range of different government departments and executive agencies in our analysis. For example, we reviewed public health documents such as Childhood Obesity and primary care documents such as NHS: Five Year Forward View. We adopted this broad approach in recognition that the determinants of poor diet and physical inactivity are affected by a range of health and societal sectors. The WHO Action Plan supports action across these different sectors which is reflected in the priority areas and policy options of our appraisal framework. The range of websites searched is listed on page 6 and the titles, authors and publication dates of all reviewed policies is provided in Additional material, Appendix 2.

7. On page 9, lines 46 – 55 findings very much reflect a top-down approach to policy setting and implementation. However, to cover needs of local populations a bottom-up approach. It is not clear how this bottom-up part of the policy setting has been considered in this analysis.

We are grateful to the reviewer for this comment. The findings the reviewer highlights describe the government departments that authored the 57 national policy documents. We agree that this illustrates the top-down approach to these policies being set. We also assessed the bottom-up policy setting adopted by each policy by including the policy options 'multistakeholder collaboration' and 'community based participatory approaches'. The policy options are defined in Additional material, Appendix 1. The results for these policy options are presented in Figures 2, 3 and 4 to show how they differ across policy types, and described in the results section of the manuscript (page 12).

8. The nature of the various documents may require a bit more detail: it is great to have a WHO endorsed document. But WHO documents are often based on model documents from its member countries, and in that way there is the possibility of no more than double-up existing UK policy documents. That way, it is not so much 'international guidance' but existing UK policy. How often did the authors find such overlap in their search?

We thank the reviewer for their comment. The aim of our study was to quantify the extent to which English national and local government policies that are relevant to maternal and child health meet the international standard set out in the WHO Action Plan. We recognise that guidance from WHO is developed from evidence and experience across member countries. In this study, the range of policy options set out in the WHO Action Plan was used as an international standard against which national English and local government action was assessed. In an attempt to clarify the purpose of the study we have reworded the aims at the end of the introduction (page 6 and below).

Page 6

There were three aims of this study. First, to identify national English government policies and local government policies, using Southampton City as a case study. Second, to analyse the extent of adoption of the Action Plan policy options in these policies. Third, to assess differences in adoption of the policy options by policy type (local/national and acts/white papers/evidence-based guidance), and across three levels of an ecological model of health.

9. Like this reviewer, readers will need a bit more details on the various categories the authors constructed. For example, what were the non-action oriented policy options (page8, lines 14 – 21)?

We are grateful to the reviewer for enabling us to improve our description of the appraisal framework. We have reworded the methods section to more clearly outline the priority areas of the appraisal framework (page 8 and below). Due to word count limitations, the definitions of each policy option are included in Additional material Appendix 1. If the reviewer and the editors felt that this table was important to include in the manuscript we could include Appendix 1 as a Table in the manuscript rather than as Additional material.

Page 8

A total of six priority areas were established for the appraisal framework. Three priority areas were action-orientated: i) maternal health, ii) infant/child health, iii) strengthening health systems. The remaining three represent public health processes necessary for high-quality policy development: iv) evidence-based strategies, v) multisectoral action, and vi) governance and accountability. Each priority area had between two and seven accompanying policy options (see Additional material, Appendix 1 for details).

10. The same is true for the findings presented on page 9: what were the policies, guidelines etc found? Keep in mind the international readership of the BMJ Open.

In Additional materials, Appendix 2 we outline the title, authors and publication date of each of the 57 national and 10 local policies reviewed in this study. The national policies are grouped according to policy type (acts and codes, white papers and evidence guidelines) and definitions of these policy types are provided in the introduction (page 4). On page 10 we direct the reader to this additional information about the 67 policies reviewed. If the reviewer and the editors felt that this table was important to include in the manuscript we could include Appendix 2 as a Table in the manuscript rather than as Additional material.

11. I am puzzled with what the authors mean by 'conflict of interest' acts and codes, in the context of this analysis. Probably giving examples could clarify this.

We are happy to clarify this term. The appraisal framework developed in this study includes a policy option managing conflicts of interest that was drawn from the overarching principle and objective 2 of the WHO Action Plan. The definition for this policy option is provided in Additional Material Appendix 1. The policy option includes explicit awareness of managing conflicts of interest with development of diet, physical activity or healthy food/physical activity environment initiatives. It does not include

reference to managing conflicts of interest related to smoking or alcohol consumption. We believe that this definition provides clear guidance for data extraction. We have included another cross-reference to Appendix 1 in the methods section of the manuscript to highlight to the reader where these definitions are provided (page 8). The findings for the conflict of interest policy option are also interpreted in the discussion section of the manuscript to provide additional explanation to the reader (page 18 and 20).

12. Could the authors expand a bit on their statement on page 16, lines 16 – 18 that evidence alone is not sufficient for policy implementation? Where are the limits of evidence and what 'best practice' is available to back this up? And to what extent is this an international best practice, transferrable to other settings, to what extent is this politics-dependent national or local business?

We have added an additional sentence to further clarify the original sentence that evidence is not adequate for policy adoption (page 17 and below). We also expand upon this point in the conclusion section to provide recommendations for future action (page 22).

Page 17

While a number of the evidence-based documents reviewed in this study did include workforce empowerment approaches, the lack of coverage in white papers indicates that evidence is not adequate for policy option adoption. Policy decision-making results from a combination of factors including the framing of an issue, political ideology, evidence of effectiveness and cost-effectiveness, timing and 'groundswell' or societal attitudes.[9, 13]

13. My final point is that in essence this is all about process – interventions to be implemented. The real interest is in its final outcome. The analysis as I stated under point 1 looks very much a search on whether all that is available is implemented. But with what avail: which guidelines, policy documents etc were just a better way of doing what already is quite well pursued in public health, and to what extent does it hold the promise of what has until now not been very successfully launched?

We thank the reviewer for allowing us to clarify the aims and importance of our study. The WHO Action Plan provides a suite of policy options that if implemented collectively will provide an international gold standard for action against non-communicable diseases (NCDs). By adopting a quantitative approach to policy analysis we were able to not only assess overall adherence to the WHO best practice but also compare the level of activity between priority areas. These analyses provide information about areas for improvement. For example, our results identified that more than three quarters of policies contained at least one of the seven maternal policy options however fewer than half included one of the four infant/child policy options. These findings indicates that greater attention could be given to adopting the recommended child policy options in the future to enhance action for prevention of NCDs relevant to dietary and physical activity behaviours. We are grateful to the reviewer for highlighting the need for us to revisit this issue in the manuscript and we have included additional information in the introduction (page 6 and below) and discussion (page 21 and below).

Page 6

Assessment of how well national and local government policies are supporting women and children to adopt healthy lifestyles is needed to identify gaps for further government action. Therefore, this study aimed to quantify the extent to which English national and local government policies that are relevant to maternal and child health meet the international standard set out in the WHO Action Plan.

Page 21

However, adopting a quantitative approach enabled assessment of overall adherence to WHO international best practice and compared the level of activity between priority areas allowing easy identification of areas for improvement.

VERSION 2 – REVIEW

REVIEWER	Anne Marie Thow University of Sydney, Australia
REVIEW RETURNED	23-Jul-2018

GENERAL COMMENTS	The authors have adequately addressed the comments.
---

REVIEWER	Chris van Weel Department of Primary and Community Care Radboud University, Nijmegen, The Netherlands also: Department of Health Services Research and Policy, Australian National University
REVIEW RETURNED	20-Jul-2018

GENERAL COMMENTS	The authors have responded in a convincing way to my earlier comments.
--